# Association of Built Environment with Physical Activity and Physical Fitness in Men and Women Living inside the City Wall of Xi’an, China

**DOI:** 10.3390/ijerph17144940

**Published:** 2020-07-09

**Authors:** Yuliang Sun, Chunzhen He, Xinxin Zhang, Wenfei Zhu

**Affiliations:** 1Department of Exercise Science, School of Physical Education, Shaanxi Normal University, Xi’an 710119, China; ysun@snnu.edu.cn (Y.S.); zhangxx9606@snnu.edu.cn (X.Z.); 2Department of Physical Education, Shanghai Ganxiang School, Shanghai 201515, China; xiaochunchun@snnu.edu.cn

**Keywords:** built environments, physical activity, physical fitness

## Abstract

This study was to investigate the association of built environment (BE) with physical activity (PA) and physical fitness of residents inside the city wall of Xi’an, one of the most historic cities in China. The cross-sectional study was conducted among 728 adults in this area. BE and PA were measured by Neighborhood Environment Walkability Scale and International Physical Activity Questionnaire, respectively. Body mass index (BMI), body fat percentage, blood pressure (BP), lung capacity, curl up, sit and reach, reaction time, balance, and grip strength were also measured. The results showed, for men, aesthetics was positively associated with total and leisure-time PA, and curl-up tests, respectively. Walking/cycling facilities were positively associated with leisure-time PA. Street connectivity was negatively associated with leisure-time PA and BMI. Residential density was positively correlated with BMI. Access to service was positively associated with lung capacity. Crime safety was negatively correlated with reaction time. For women, residential density was negatively associated with transportation PA and BP. Street connectivity was positively associated with curl-up test. The results suggest some BE attributes are positively related to PA and physical fitness in this population. Creating more PA-supportive BEs is recommended in this historic area along with urban conservation.

## 1. Introduction

With the progress of economic development, the health consciousness of residents in China has been constantly improving. However, studies have shown that most Chinese adults do not have enough time and place to exercise, which could be harmful to their health [1,2]. Research also shows that a sedentary lifestyle is one of the most important reasons affecting health [3,4]. Less physical activity (PA) is associated with increasing incidence of chronic diseases, and the risk of chronic disease is higher in sedentary people [1,5,6].

Built environments (BEs) are one of the most important factors affecting PA, sedentary behaviors and health [7,8,9]. For example, it has been proven that a positive association exists between residential density and transportation-related walking [10]. There are also positive associations between land-use-mix and any bouts of walking/cycling for transportation PA [11]. Some researchers have confirmed street lighting near the residence, the planning of sidewalks and vehicle ways, and other road facilities are positively related to residents’ PA levels [12,13]. Additionally, public order in the community, the artificial or natural aesthetic factors of the environment, and the construction of public entertainment places are also positively associated with residents’ PA [14].

The relationship between BE and PA has been confirmed in different countries and regions. For example, some studies found that higher scores for the BE were associated with higher levels of PA in residents of Brazil [15,16,17]. There are also researchers from Poland and the Czech Republic that found higher environmental scores are related with higher PA levels [18]. Simultaneously, the surveys conducted in Shanghai and Hangzhou in China also concluded that the BE is positively associated with PA level of city residents in East China [19,20]. However, those less developed regions, such as the northwest areas of China, have not been well studied yet. The relationship between BE and PA needs to be verified in different areas with diverse historical backgrounds, cultures, urban planning and city designs.

Physical fitness can predict changes in functional, psychological and social health in young and older populations [6]. Physical fitness is associated with many health benefits, such as lower cardiovascular disease risk, increasing bone mineral density, improving glucose tolerance and healthy aging [6]. Research has shown a more activity-supportive BE was associated with better physical fitness [21], lower blood pressure [22], and lower risks of obesity [23]. However, fewer studies have examined the association between BE and physical fitness, especially cardiorespiratory fitness, muscular strength, flexibility and speed/agility. Most previous studies were focused on children and adolescents [21]. No study has analyzed the association of BE with physical fitness of adults living in China.

Xi’an, located in central-northwest China, has more than 3000 years of history, including over 1100 years as the capital city of ancient dynasties. It is also the oldest prefecture capital, and one of the four great ancient capitals. The city walls of Xi’an, built as a military defense system in the 14th century, represent one of the oldest, largest and best preserved Chinese city walls. It is on the tentative list of United Nations Educational, Scientific and Cultural Organization as the World Heritage Sites under the title “City Walls of the Ming and Qing Dynasties”. Since 2009, the city walls and the areas around have been protected by the cultural relics, which are different from other modern cities. New building construction or major social events inside the city wall must be approved by the government. Additionally, the Xi’an Muslim Quarter, which is a relatively large Hui community (non-Uyghur Chinese Muslims), is located inside the city wall. The Muslim Quarter has existed for centuries since the Ming Dynasty and covers several blocks inhabited by over 50,000 Muslims. Therefore, the area inside the city wall is a very well-known tourism site which is famous for its long history, diverse culture and feature food. To date, no study has investigated the BE and its relationship with PA, and physical fitness in such a historic area with traditional urban landscape.

Thus, our study aimed to investigate the BE, PA and physical fitness of adults living inside the city wall of Xi’an. In our research, we hypothesized that there was a positive correlation of BE with residents’ PA and physical fitness. More activity-supportive BE (higher scores in BE attributes) were significantly related with increasing levels of PA, lower risk of obesity, lower blood pressure (BP) and increasing levels of physical fitness, respectively. Our analyses have provided new information for the association of BE with residents’ PA and physical fitness in a city like Xi’an, an ancient capital in China, and established a relevant theoretical model for improving the urban conservation while fulfilled the basic living, cultural and active living needs of local communities.

## 2. Materials and Methods

### 2.1. Sample and Study Design

The study area was the communities inside the city walls of Xi’an in the northwest of China. Xi’an is one of the few cities in China where the imposing old city walls still stand. Built in 1370 during the Ming Dynasty, the magnificent 12 m-high walls form a rectangle with a perimeter of 14 km. The opening of gates through the city wall helps to maximize its preservation while meeting the needs for urban transportation. Currently, the city wall has been transformed into an urban public space facilitated by the surrounding building into a city-dominated landscape. Xi’an was also the first city in China where Muslims started arrived since 651, during the Tang Dynasty. Xi’an’s Muslim community is located inside the city wall, and the population is estimated to be around 50,000.

In this study, 728 residents from 15 different communities inside the city wall of Xi’an were recruited through community hospitals for our study. Participants were those aged 18 years and older, with normal reading and speech abilities, and no cognitive impairment diagnosed by doctors. The average age of participants was 46.9 ± 13.5 years old. All participants in this study have signed written consents to participate. The study was approved by the ethics committee of Shaanxi Normal University. A structured questionnaire was designed and issued in September 2018. Additionally, height, weight, body fat percentage (BFP), BP, lung capacity, curl up, sit and reach, reaction time, balance, and grip strength were tested in our laboratory.

### 2.2. Measures

Participants were asked to fill the questionnaire about their individual characteristics, including sex, age, address, employment, history of chronic disease, family income, et cetera. The Chinese version of Neighborhood Environment Walkability Scale—Abbreviated (NEWS-A) and International Physical Activity Questionnaire—long version (IPAQ-L) were completed in the questionnaire survey. The questionnaires were filled by our research team members through one-to-one interviews to ensure the quality.

There were eight dimensions in the NEWS-A, including residential density (score range: 173–594), land use mix diversity (range: 1–5), access to services (range: 1–4), street connectivity (range: 1–4), walking and cycling facilities (range: 1–4), aesthetics (range: 1–4), traffic safety (range: 1–4), and crime safety (range: 1–4) [24]. IPAQ-L was used to evaluate PA levels during the past seven days, including questions for work PA, transportation PA, household PA, and leisure-time PA.

Body height and weight were measured with the participants taking off their shoes and standing on the scale platform. The participants were asked to stand in front of the vertical scale and face forward maintaining a horizontal focus. A bar from the top of the scale was pushed down to the top of the participant’s head. The height and weight were recorded by the scale.

Body composition tests were performed by Inbody 230 (IT: Biospace Co Ltd., Seoul, South Korea). The participants removed all metal objects and took off their shoes and socks during the test. The metal area of the machine was in direct contact with the skin.

BP was measured by trained staff using an automated device (IT: Omron Healthcare Europe BV, Hoofddorp, The Netherlands) with appropriate mid-upper-arm circumference cuff size. BP measurements were taken on the left arm of each participant in a sitting position after being at rest for at least two minutes. Two readings were made three to four minutes apart and the averaged readings were recorded.

Lung capacity was measured by electron pneumonometer (WCS-1000, Donghong Technology Development Center, Beijing, China). The largest exhalation volume was recorded of three attempts. Participants were reminded not to blow too hard, not to stop halfway, and not to block the air outlet.

Flexibility was tested by sit and reach test (Digital electronic fleximeter, Donghong Technology Development Center, Beijing, China). Participants were told to straighten their legs after taking off their shoes, and their upper limbs are straightened to push the card mark at a uniform speed. Participants were reminded to keep the continuity of the movements. Both hands should remain at the same level, not one reaching further forward than the other. After some practice reaches, the participants reached out and held that position for one to two seconds while the distance was recorded.

Curl up test was used to test the strength and endurance of the abdominal muscles. It required the participants to perform as many curl-ups as possible in one minute. The participant began by lying on their back, knees bent at approximately 140 degrees, feet flat on the floor, legs slightly apart, and arms straight and parallel to the trunk with palms of the hands resting on the mat. The fingers were stretched out and the head was in contact with the mat. The measuring strip was placed on the mat under their legs so that their fingertips were just resting on the nearest edge of the measuring strip. The feet cannot be held or rest against an object. Keeping their heels in contact with the mat, the participant curled up slowly, sliding their fingers across the measuring strip until the fingertips reached the other side, then curled back down until their head touched the mat. Movement should be smooth and at the cadence of 20 curl-ups per minute (1 curl-up every 3 s).

Reaction time was conducted to assess the neurofunction and muscle response. The participants pressed the red button to initiate a signal so that the system (WCS-100, Donghong Technology Development Center, Beijing, China) returned a signal by lighting one of five green buttons. The participant then pressed the lit green button as quickly as possible, and the time between the red and green button pressing was measured. A total of five measurements were taken and the average time was recorded.

Grip strength was tested by hand dynamometer (WCS-100, Donghong Technology Development Center, Beijing, China). Participants kept their feet apart in an upright position and their arms were drooped naturally. They were required to squeeze the dynamometer twice with the dominant hand and take the maximum value. Their hands were not allowed to touch their legs.

One-foot standing test with closed eyes was used to evaluate their balancing ability. The participants were asked to stand on the sensor with their feet together and their arms at their sides. When instructed, they closed their eyes and lifted one foot (either one) off the ground until they swayed while balancing, used arms to balance, hopped to maintain balance, or put the lifted foot down. The participants repeated the assessment three times, and the longest time was recorded.

The equipment, protocol, and testing items were calibrated, and the same models of the equipment were used for every measurement. For comparison, all equipment and protocols matched those used by the General Administration of Sport of China.

### 2.3. Statistical Analysis

Independent variables were the eight dimensions of environmental variables (residential density, land use mix diversity, access to services, street connectivity, walking/cycling facilities, aesthetics, pedestrian/traffic safety, and crime safety), as well as the total scores of BE. The dependent variables included PA, body mass index (BMI), BFP, BP, lung capacity, curl up, sit and reach, reaction time, balance, and grip strength. Descriptive statistics were used to describe sample characteristics. Significant differences were found in most dependent variables between different sex groups. Thus, all the analyses were done within each sex group. A mixed regression model was used to analyze the association of BE with PA (total, transportation, and leisure-time moderate-to-vigorous physical activity (MVPA)) and physical fitness of the participants, with age, work status, ethnic group, number of family numbers, and marital status as the control variables. In the regression model for the relationship between BE and physical fitness (including BMI, BFP, systolic blood pressure (SBP), diastolic blood pressure (DBP), lung capacity, curl up, sit and reach, reaction time, balance, and grip strength), age, work status, ethnic group, number of family numbers, marital status, and total MVPA were controlled. Statistical significance was set at *p* < 0.05. All the analyses were conducted by SPSS 24.0 software (SPSS Inc., Chicago, IL, USA).

## 3. Results

Table 1 presents the characteristics of the participants. In our study, 59.5% of the participants were women; 30.6% aged 18–40 years, 41.2% aged 41–60 years, and 28.2% over 60 years. Most of them have jobs (93.3%), three to four family members (63.6%), and 80.4% were married. About 25.3% of the participants were from the Hui (non-Uyghur Chinese Muslim) community.

The BE scores in each dimension were relatively medium to high according to the ranges of the scores (all above the median scores in each dimension) in this studied area. Significant differences existed in the scores of residential density and access to services between women and men (*p* < 0.05). The MET values of different types of PA for men and women are shown in Table 2. Men spent the highest proportion of their time in leisure time PA, and the lowest proportion of their time in household PA. In the women groups, the proportion of household PA was the largest, and that of transportation PA was the smallest. Significant differences were found in household PA, transport PA%, household PA%, BMI, BFP, grip strength, lung capacity, sit, and reach between men and women (*p* < 0.05). Women had significantly more household PA, household PA%, and less leisure-time PA%. BMI was significantly higher in men while BFP was significantly higher in women (*p* < 0.05). Men performed significantly better in grip strength, lung capacity, but worse in sit and reach than women (*p* < 0.06). Similar analyses have been done for different ethnic groups (Han or Hui groups). No significance has been found between Han and Hui groups (*p* > 0.05).

The correlations between BE and PA are shown in Table 3. Aesthetics was positively associated with total and leisure-time PA in men (*p* < 0.05). Street connectivity was negatively associated with leisure-time PA in men (*p* < 0.05). Walking/cycling facilities were positively associated with leisure-time PA in men (*p* < 0.05). Residential density was negatively associated with transportation PA in women (*p* < 0.05). No significant correlation was found between other BE features and transportation/leisure-time PA (*p* > 0.05). Work-related PA and domestic and gardening PA were also analyzed in our study. However, no significant relationship has been found. Therefore, they were no longer presented in the table.

The correlations between BE and BMI, BFP, SBP, and DBP are presented in Table 4. For men, residential density was positively correlated with BMI (*p* < 0.05). Street connectivity was negatively associated with BMI (*p* < 0.05). For the women, residential density was negatively associated with SBP and DBP (*p* < 0.05). No significant association was found in other measurements (*p* > 0.05).

The correlations between BE and physical fitness are shown in Table 5. For men, access to services was positively associated with lung capacity. Aesthetics was positively correlated with curl-up test (*p* < 0.05). Access to services, traffic safety, and land use mix diversity were negatively associated with sit and reach test, respectively (*p* < 0.05). Crime safety was negatively correlated with reaction time (*p* < 0.05). For women, street connectivity was positively associated with curl-up test (*p* < 0.05). No significant association was found in other measurements (*p* > 0.05). Due to the space limits, the testing results of grip strength and standing test with closed eyes were not listed in Table 5.

Figure 1 summarizes the results of the association of BE with PA and physical fitness for men and women, respectively. Higher rated BE attributes were associated with higher levels of PA (leisure-time PA for men and transportation PA for women) and better performance in some physical fitness tests among residents living in this historic area, especially for men. BE attributes like residential density and street connectivity were better-related to BMI and BP, while access to services, aesthetics, and crime safety were better-correlated with some physical fitness variables.

## 4. Discussion

In the present study, questionnaire surveys and objective physical fitness tests were conducted to identify characteristics and relationships of BE with PA and physical fitness among adults living inside the city wall of Xi’an. In recent years, Xi’an has become an important economic, cultural, industrial, and educational center in the central-northwest region of China. In this study, we found that the BE in this area is favorable for PA and physical fitness because the modern city can provide convenient facilities and resources. Men mainly engaged in leisure-time PA, while women mainly engaged in household PA. No significant difference has been found in PA and physical fitness between Han and Hui ethnic groups, indicating their lifestyles may be similar after long-term cultural communication with each other. For most measures, more activity-supportive BE attributes were associated with higher levels of PA and better physical fitness among residents living inside the city walls in Xi’an, especially for men. Our study is one of the first to investigate the association of BE inside the city wall of Xi’an with PA and physical fitness of residents living in such a historic area with diverse cultural context. It provides urban planners with potential priorities for healthier neighborhoods in local communities while improving the urban conservation in historic Chinese cities.

In our analysis of the relationship between BE and PA, BE attributes were related to total PA, transportation PA, and leisure time PA. This was consistent with the socio-ecological model [25] and several previous studies [26,27,28,29,30,31]. For example, aesthetics was positively associated with total and leisure-time PA in men, indicating aesthetics can motivate residents to engage in more PA [32] and men were more likely to conduct leisure-time activities in aesthetically pleasing spaces [33,34,35]. Street connectivity was negatively associated with leisure-time PA in men. However, most previous studies [12,13,14] reported that street connectivity is positively associated with PA in men and women. The reason for the different results in our study may be that the tourism industry inside the city wall is well developed and there are many floating populations. The residents may avoid traveling during rush hour, which reduces the PA of leisure-time. Our study has also shown that walking/cycling facilities were positively associated with leisure-time PA. This was similar to previous researches [36,37]. With the development of China, there is increasing ownership of private vehicles in big cities like Xi’an. Walking/cycling facilities provide new choices so that people can travel faster in rush hour if their destination is not very far away. Therefore, walking/cycling facilities may have a positive influence on transportation or leisure-time PA. For the women group, we found that residential density was negatively associated with transportation PA. However, previous studies have found that the possibility of using transportation is higher in denser, more diversified urbanized settings [38,39], compact neighborhoods usually contribute to health-related physical activities [38,40]. The reason for the contradictory results in our study may be that larger residential density leads to heavier traffic, which was negatively associated with PA [21]. Express delivery and take-out food are very popular and convenient in China, and the expense is affordable. People in the denser city tend to choose shopping online to avoid the traffic in the real world, and women may be more sensitive to those BE attributes. Because both BE and PA were measured by self-reported methods, the relationship may suffer from subjective bias. Further investigation is needed to figure out the issues.

In our analysis of the association of BE with BMI and BP, residential density was positively correlated with BMI, and street connectivity was negatively associated with BMI in men. BE was not associated with the adiposity of women. Previous research has shown that a more activity-supportive BE and a higher level of PA were related to a lower probability of obesity [27,41,42,43,44,45]. However, our results were mixed in a different dimension of BE between men and women. We conducted further field surveys of residential areas in our study and found the number of fast-food restaurants, specialty restaurants, and snack-bars in the survey area were more than that in other residential areas. The number of restaurants and convenience stores near residential areas has been shown to have an impact on obesity [46,47,48]. Furthermore, we considered that the local food culture and the local eating habits in the city of Xi’an are for oily and spicy foods, with noodles as the main food. Therefore, the reasons for the research results were not only related to the construction environment but also the eating habits, which were also one of the significant factors affecting the obesity of residents. Meanwhile, for the women group, our studies found that residential density was negatively associated with SBP and DBP. Previous studies have also confirmed the relationship between BE and BP. With more public places nearby, such as bus stations, subway stations, et cetera, residents may prefer walking or riding bicycles, accumulate more PA, and thus the risk of hypertension may be lower [41,49]. However, residential density was not associated with PA in women according to our analyses. There may be some other median factors in the relationship between BE and BP, such as food intake. Further investigation is needed in the larger population.

The association between BE and physical fitness was mixed between men and women. For men, access to services was positively associated with lung capacity. It has been shown that higher levels of access to services are significantly associated with PA [42,50,51], thereby possibly increasing men’s lung capacity. Interestingly, we also found in our studies that access to services, traffic safety, and land use mix diversity were negatively associated with sit and reach tests, respectively. This may be related to some other factors, such as the pre-activity warm-up program [52], and the specific reason needs further investigation. In our studies, crime safety was negatively correlated with reaction time. This is similar to previous studies’ results that lower criminal activity will increase the chance of residents going out to exercise [25,53] so indirectly improving the reaction time. Our research also analyzed the relationship between BE and the curl-up test, and the results indicated that aesthetics was positively associated with the curl-up test in men, and street connectivity was positively associated with curl-up test in women. No previous researches were found focusing on the relationship between these two. We speculate that this may be related to the activity habits of different groups of people. A higher activity-supportive BE may support people in developing long-term exercise habits, helping them to improve their physical fitness in different aspects. More studies are necessary to confirm the exact association between BE and physical fitness in diverse populations.

The strengths of our study included: firstly, the selected research area was representative, and it was also the first time the correlation of the BE of a historic city with residents’ PA and physical fitness was analyzed in China. Secondly, the physical fitness, including lung capacity, curl up, sit and reach, reaction time, balance, and grip strength, as well as BMI, BFP, and BP were objectively evaluated in our study, providing detailed information about the aerobic capacity, muscle strength, flexibility, balance, and attention in this population. Finally, the NEWS-A and IPAQ-L are internationally recognized and have high reliability and validity, which can effectively provide the data needed for this study. The quality of questionnaire filling was high, which avoided the deviation of data caused by a different understanding of personal subjective factors. The present study had several limitations. Firstly, although we avoided deviations as much as possible in the process, the measurements of BE and PA were both evaluated by self-reported tools. Same source bias may result in associations between independent and dependent variables. However, it still provided some new information about the levels of BE and PA and their possible relationships for men and women living inside the city wall of Xi’an. Fortunately, the relationship between the BE and physical fitness was more convincing because objective measurements were conducted for the physical fitness tests. Secondly, it was a cross-sectional study, and causal relationships cannot be inferred. Third, the pattern of the results was mixed because the association for different sexes, aspects of BE, and physical fitness varied and needs to be discussed respectively. Finally, social and psychological factors were not studied in this study. Some research has shown that social support and self-efficacy may also be correlated with residents’ PA and health status. In the future, objectively measured BE and PA, such as by GIS and accelerometer, should be used to investigate this area. Longitudinal or experimental studies considering more personal deviation will help to figure out the impact of BE on PA and physical fitness.

## 5. Conclusions

In general, higher rated BE attributes were associated with higher levels of PA and better physical fitness among residents living in this historic area, especially for men. Characteristics like residential density and street connectivity were better-related to BMI and BP, while access to services, aesthetics, and crime safety were better-correlated with some physical fitness variables. The study suggests designing activity-supportive BEs should be a higher health priority to increase PA and enhance physical health of local residents. Further research is necessary to dig out the exact impact of BE on PA and physical fitness to promote wellbeing of city residents while improving the urban landscape of those historic cities.

## Figures and Tables

**Figure 1 ijerph-17-04940-f001:**
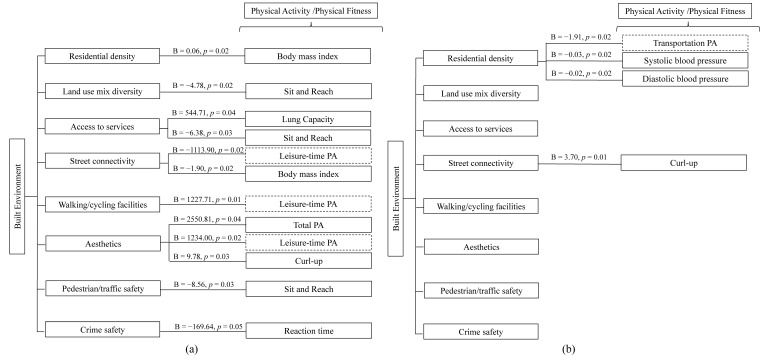
Summary of association between built environment attributes and physical activity/ physical fitness variables in men and women living inside the city wall of Xi’an. PA indicates physical activity. (**a**) Is for men and (**b**) is for women. Solid lines indicate the built environment attributes and physical fitness variables, dashed lines indicate the physical activity variables.

**Table 1 ijerph-17-04940-t001:** Demographics of the participants (*n* = 728).

Demographic Variable	*n*	%
Sex		
Men	295	40.5
Women	433	59.5
Age		
18–40 years	223	30.6
41–60 years	300	41.2
≥60 years	205	28.2
Work status		
Employed	679	93.3
Unemployed	49	6.7
Ethnic groups		
Han	544	74.7
Hui (non-Uyghur Chinese Muslim)	184	25.3
Other	0	0
Number of family members		
1–2	120	16.5
3–4	463	63.6
≥5	145	19.9
Marital status		
Married	585	80.4
Unmarried	143	19.6

**Table 2 ijerph-17-04940-t002:** Description of BE, PA, and physical fitness in different sex groups (mean ± SD).

Variable	Men	Women
Residential density	391.9 ± 106.7	453.7 ± 145.4 *
Land use mix diversity	2.7 ± 0.7	2.8 ± 0.7
Access to services	3.3 ± 0.5	3.1 ± 0.5 *
Street connectivity	3.0 ± 0.6	2.9 ± 0.7
Walking/cycling facilities	2.7 ± 0.6	2.8 ± 0.6
Aesthetics	2.1 ± 0.6	2.3 ± 0.7
Pedestrian/traffic safety	2.8 ± 0.4	2.8 ± 0.4
Crime safety	3.4 ± 0.5	3.4 ± 0.7
Total BE score	398.8 ± 128.6	441.9 ± 159.7 *
Transportation PA (MET·min/week)	937.7 ± 767.6	1013.9 ± 1046.2
Leisure-time PA (MET·min/week)	1803.7 ± 1606.0	1249.4 ± 1937.4
Working PA (MET·min/week)	1778.6 ± 3383.3	2276.9 ± 5411.0
Household PA (MET·min/week)	754.8 ± 713.2	1979.9 ± 2491.2 *
Total MVPA (MET·min/week)	5274.8 ± 3955.7	6493.7 ± 6530.6
Transportation PA%	22.1 ± 18.5	21.5 ± 19.2
Leisure-time PA%	33.9 ± 21.5	22.0 ± 23.2 *
Working PA%	23.3 ± 28.5	23.1 ± 29.8
Household PA%	20.8 ± 24.9	33.7 ± 27.1 *
Body mass index (kg·m^−2^)	23.9 ± 2.8	22.4 ± 2.4 *
Body fat percentage (%)	22.3 ± 5.4	28.9 ± 4.8 *
Systolic blood pressure (mmHg)	122.3 ± 15.7	116.9 ± 17.2
Diastolic blood pressure (mmHg)	78.7 ± 10.4	74.0 ± 11.5
Curl up (count/min)	22.7 ± 11.0	15.1 ± 9.0
Grip strength (kg)	39.1 ± 7.9	27.3 ± 4.6 *
Standing test with closed eyes (s)	20.7 ± 24.6	19.7 ± 18.8
Lung capacity (ml)	3386.5 ± 875.7	2411.4 ± 630.3 *
Sit and reach (cm)	0.6 ± 7.6	8.7 ± 9.5 *
Reaction time (ms)	452.9 ± 241.3	411.8 ± 146.3

Note: BE indicates built environment, PA indicates physical activity, MVPA indicates moderate-to-vigorous physical activity, * indicates significant differences between men and women (*p* < 0.05).

**Table 3 ijerph-17-04940-t003:** Association ^a^ between BE and transportation PA, leisure-time PA.

BE Variable	Total PA			Transportation PA			Leisure-Time PA		
	B	SE	*p*	B	SE	*p*	B	SE	*p*
**Men**									
Residential density	−4.37	8.08	0.59	0.09	1.57	0.96	4.29	3.20	0.19
Land use mix diversity	−1403.45	1075.01	0.20	32.15	214.43	0.89	129.25	449.13	0.78
Access to services	−1497.98	1573.49	0.35	84.36	309.32	0.79	89.42	649.24	0.89
Street connectivity	−1152.67	1222.34	0.35	−282.34	234.54	0.24	−1113.90	458.34	0.02 *
Walking/cycling facilities	1676.11	1211.63	0.18	306.07	235.63	0.21	1227.71	452.76	0.01 *
Aesthetics	2550.81	1215.55	0.04 *	−308.27	246.54	0.22	1234.00	477.28	0.02 *
Pedestrian/traffic safety	3094.94	1869.91	0.11	−130.55	378.77	0.73	1122.48	767.11	0.16
Crime safety	−2493.02	1398.69	0.09	−259.24	281.70	0.37	764.06	581.91	0.20
Total	−6.48	5.79	0.27	−0.86	1.14	0.46	2.57	2.36	0.28
**Women**									
Residential density	3.40	5.23	0.52	−1.91	0.81	0.02 *	−0.42	1.53	0.79
Land use mix diversity	−1050.12	1092.21	0.34	110.86	175.36	0.53	−195.92	320.03	0.54
Access to services	−1043.99	1460.74	0.48	98.85	234.26	0.09	470.87	424.78	0.27
Street connectivity	−864.26	979.40	0.38	153.92	156.58	0.70	−289.13	285.61	0.31
Walking/cycling facilities	1052.92	1344.42	0.44	82.77	215.78	0.33	−95.75	393.94	0.81
Aesthetics	−526.91	1058.59	0.63	−21.03	169.66	0.90	−66.76	309.51	0.83
Pedestrian/traffic safety	2023.40	1663.77	0.23	182.13	267.91	0.50	−101.84	490.08	0.84
Crime safety	−1007.15	1041.58	0.34	77.19	167.43	0.65	−223.29	304.91	0.47
Total	1.00	4.36	0.82	−1.05	0.69	0.13	0.10	1.27	0.94

Note: BE indicates built environment; PA indicates physical activity; MVPA indicates moderate-to-vigorous physical activity; * indicates significant correlation (*p* < 0.05); ^a^ indicates the model controlled for age, work status, ethnic group, number of family numbers, and marital status.

**Table 4 ijerph-17-04940-t004:** Association ^b^ between BE and BMI, BFP, SBP, and DBP.

BE Variable	BMI			BFP			SBP			DBP		
	B	SE	*p*	B	SE	*p*	B	SE	*p*	B	SE	*p*
**Men**												
Residential density	0.06	0.02	0.02 *	−0.01	0.01	0.32	0.05	0.03	0.15	0.01	0.02	0.68
Land use mix diversity	−0.01	0.71	0.99	0.46	1.56	0.77	−4.83	4.21	0.26	−1.19	2.76	0.67
Access to services	1.52	0.95	0.12	0.90	2.21	0.69	−8.01	5.90	0.19	−4.01	3.84	0.31
Street connectivity	−1.90	0.79	0.02 *	−3.50	1.82	0.07	−5.72	5.25	0.29	−3.27	3.38	0.34
Walking/cycling facilities	−0.55	0.81	0.51	−2.24	1.77	0.22	3.61	3.40	0.29	2.36	3.18	0.47
Aesthetics	−0.72	0.99	0.47	0.18	2.21	0.94	1.65	6.10	0.79	−1.56	3.90	0.69
Pedestrian/traffic safety	−1.05	1.27	0.42	−2.24	2.83	0.44	−6.80	7.78	0.39	−2.65	5.04	0.60
Crime safety	0.13	0.96	0.90	0.31	2.13	0.89	1.30	5.88	0.83	4.27	3.68	0.26
Total	0.00	0.00	0.89	−0.01	0.01	0.27	0.02	0.02	0.42	0.01	0.02	0.74
**Women**												
Residential density	−0.01	0.01	0.61	0.00	0.00	0.26	−0.03	0.01	0.02 *	−0.02	0.01	0.02 *
Land use mix diversity	0.28	0.41	0.50	1.34	0.80	0.10	0.24	2.78	0.93	−0.54	1.89	0.78
Access to services	−0.12	0.56	0.84	−0.62	1.11	0.58	−3.85	3.78	0.31	0.89	2.58	0.73
Street connectivity	−0.05	0.37	0.89	−0.48	0.74	0.52	0.90	2.54	0.73	0.90	1.73	0.60
Walking/cycling facilities	0.44	0.50	0.38	1.09	0.99	0.28	5.81	4.89	0.25	2.05	2.32	0.38
Aesthetics	0.19	0.40	0.64	0.82	0.78	0.30	0.71	2.70	0.79	−0.09	1.84	0.96
Pedestrian/traffic safety	1.14	0.61	0.07	2.30	1.22	0.06	0.79	4.26	0.85	4.90	2.84	0.09
Crime safety	−0.09	0.41	0.83	−0.89	0.81	0.27	−1.42	2.78	0.61	−0.81	1.89	0.67
Total	0.00	0.00	0.09	0.00	0.00	0.29	−0.02	0.01	0.10	−0.01	0.01	0.08

Note: BE indicates built environment; BMI indicates body mass index; BFP indicates body fat percentage; SBP indicates systolic blood pressure; DBP indicates diastolic blood pressure; PA indicates physical activity; MVPA indicates moderate-to-vigorous physical activity; * indicates significance (*p* < 0.05); ^b^ indicates the model controlled for age, work status, ethnic group, number of family numbers, marital status, and total moderate-to-vigorous physical activity.

**Table 5 ijerph-17-04940-t005:** Association ^c^ between BE and lung capacity, curl-up, sit and reach, and reaction time in different PA groups.

BE Variable	Lung Capacity			Curl-Up			Sit and Reach			Reaction Time		
	B	SE	*p*	B	SE	*p*	B	SE	*p*	B	SE	*p*
**Men**												
Residential density	−0.96	1.64	0.56	−0.03	0.02	0.28	0.01	0.02	0.43	−0.54	0.48	0.27
Land use mix diversity	203.73	212.91	0.35	−1.69	3.19	0.60	−4.78	2.00	0.02 *	−72.20	62.49	0.26
Access to services	544.71	287.10	0.04	3.98	4.47	0.38	−6.38	2.86	0.03 *	1.80	90.61	0.98
Street connectivity	413.68	256.46	0.12	−4.22	3.90	0.29	1.48	2.73	0.59	−104.38	76.89	0.19
Walking/cycling facilities	18.09	251.60	0.94	4.95	3.60	0.18	2.11	2.53	0.41	-8.32	74.41	0.91
Aesthetics	−179.28	304.25	0.56	9.78	4.11	0.03 *	−2.01	3.10	0.52	48.75	90.09	0.59
Pedestrian/traffic safety	462.68	385.47	0.24	−2.30	5.85	0.70	−8.56	3.67	0.03 *	29.89	116.99	0.80
Crime safety	−517.80	276.95	0.07	0.75	4.37	0.87	−0.27	3.00	0.93	−169.64	80.66	0.05 *
Total	1.30	1.18	0.28	0.002	0.02	0.93	0.01	0.01	0.65	−0.32	0.35	0.37
**Women**												
Residential density	0.38	0.47	0.42	0.01	0.01	0.26	0.01	0.01	0.40	0.05	0.12	0.66
Land use mix diversity	23.89	97.65	0.81	−1.65	1.45	0.26	0.25	1.62	0.88	−12.01	24.08	0.62
Access to services	−163.02	132.42	0.22	−2.06	1.98	0.30	0.15	2.22	0.95	−19.04	32.93	0.57
Street connectivity	−135.37	88.12	0.13	3.70	1.27	0.01 *	1.99	1.47	0.18	7.54	22.06	0.73
Walking/cycling facilities	−97.32	119.79	0.42	−1.83	1.79	0.31	2.53	1.98	0.21	4.34	29.69	0.88
Aesthetics	−17.87	94.92	0.85	−1.99	1.40	0.16	−0.24	1.58	0.88	−3.30	23.44	0.89
Pedestrian/traffic safety	−70.88	149.48	0.64	2.43	2.22	0.28	−0.95	2.48	0.70	1.09	36.96	0.98
Crime safety	−22.94	97.86	0.82	−0.49	1.46	0.74	1.35	1.62	0.41	1.10	24.17	0.96
Total	−0.04	0.39	0.92	0.01	0.01	0.16	0.01	0.01	0.25	−0.18	0.09	0.06

Note: BE indicates built environment; PA indicates physical activity; MVPA indicates moderate-to-vigorous physical activity; * indicates significant correlation (*p* < 0.05); ^c^ indicates the model controlled for age, work status, ethnic group, number of family numbers, marital status, and total moderate-to-vigorous physical activity.

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
