# Peer review of "Association of Built Environment with Physical Activity and Physical Fitness in Men and Women Living inside the City Wall of Xi’an, China"

_ijerph, 2020, doi:10.3390/ijerph17144940_

Round 1

Reviewer 1 Report

While the authors have made great efforts to improve the first submission (specially by the introduction of physical condition variables, there are some important issues that they still need to address before considering this manuscript for publication at IJERPH:

  • If the novelty of the study in the association of BE-physical condition, and the association between BE-Physical activity might be biased by the same source bias, I don’t understand why the analysis of physical activity are necessary
  • The authors need to adjust for some multiple comparison test. It is difficult to see a clear pattern in the results, and they report as main findings only the ones that they found significant.
  • There is a need to further discuss how the different aspects of the BE impact fitness. Characteristics like walkability might be better-related to BMI or blood pressure, while the presence of exercise facilities might be better-correlated with proper fitness outcomes.

Author Response

Reviewer 1

While the authors have made great efforts to improve the first submission (specially by the introduction of physical condition variables, there are some important issues that they still need to address before considering this manuscript for publication at IJERPH:

If the novelty of the study in the association of BE-physical condition, and the association between BE-Physical activity might be biased by the same source bias, I don’t understand why the analysis of physical activity are necessary.

Response: Thank you for the comments. We admitted that the BE and PA may be biased by the same source of bias and have mentioned the point in the limitation. However, PA is a variable influenced by BE directly, and the inclusion of PA still can provide some new information about the possible association between BE and PA in men and women living inside the City Wall of Xi’an. The current analyses of PA showed BE attributes were related to total PA, transportation PA and leisure time PA, suggesting designing activity-supportive BEs should be a higher health priority to increase PA and enhance physical health of local residents. Future research using GIS and accelerometer is recommended in our discussion. To clarify this point, we discussed it a little more in the last part of the Discussion (Page 10, Line 331-333).

The authors need to adjust for some multiple comparison test. It is difficult to see a clear pattern in the results, and they report as main findings only the ones that they found significant.

Response: Thank you for your comments. The associations between BE and PA/physical fitness were mixed between men and women. Analysis for the total BE score has been conducted, and no relationship has been found. Thus, the associations of different dimensions were analyzed respectively, depending on sex, BE attributes and PA/physical fitness variables. We have tried different methods for adjustment, but statistical analyses were not able to change the situation. The mixed pattern in our results may due to the cross-sectional design, and self-reported tool bias. The limitation has been talked in the discussion (Page 10, Line 335-337), and further investigation is needed to better understand the actual relationship of BE with PA and physical fitness in this population living in the area. We will conduct similar research to improve this aspect in the future.

There is a need to further discuss how the different aspects of the BE impact fitness. Characteristics like walkability might be better-related to BMI or blood pressure, while the presence of exercise facilities might be better-correlated with proper fitness outcomes.

Response: Thanks for the comment. We have modified our manuscript accordingly (Page 10, Line 335-337; Page 10, Line 344-346). However, it was hard to generalize the pattern of the results because the association for different sexes, BE attributes, and physical fitness were different and need to be discussed respectively. Additionally, fewer studies existed for us to compare the results with, especially for those fitness tests like lung capacity, curl-up, sit and reach, and reaction time. Further investigation is necessary to improve our understanding about how the different aspects of the BE impact fitness.

Reviewer 2 Report

This paper analyze the of built environment with physical activity and physical fitness of residents inside the City Wall of Xi'an, one of the most historic cities in China.
The work is partially interesting. The authors have modified the manuscript according to provided suggestions. Now the research project is appropriate, the conceptional structure is well organized.
The reviewer wonders only about the age of participants.
How can authors explain the change in age with the same average?
It was:
Participants in this study were those aged between 15 and 69 years old, with normal reading and speech abilities, and no cognitive impairment diagnosed by doctors. The average age of participants was 46.6± 13.8 years old
Now is it:
Participants were those aged equal or above 18 years old, with normal reading and speech abilities, and no cognitive impairment diagnosed by doctors. The average age of participants was 46.6± 13.8 years old.

Author Response

Reviewer 2

This paper analyzed the of built environment with physical activity and physical fitness of residents inside the City Wall of Xi'an, one of the most historic cities in China.

The work is partially interesting. The authors have modified the manuscript according to provided suggestions. Now the research project is appropriate, the conceptional structure is well organized.

The reviewer wonders only about the age of participants.

How can authors explain the change in age with the same average?

It was:

Participants in this study were those aged between 15 and 69 years old, with normal reading and speech abilities, and no cognitive impairment diagnosed by doctors. The average age of participants was 46.6± 13.8 years old

Now is it:

Participants were those aged equal or above 18 years old, with normal reading and speech abilities, and no cognitive impairment diagnosed by doctors. The average age of participants was 46.6± 13.8 years old.

Response: Thank you for the comments. According to the reviewer’s suggestion, participants aged less than 18 years old were excluded in the analyses. The average age has been modified in the text as “46.9± 13.5 years old” (Page 3, Line 98). Also, the demographic information has been edited in Table 1.

Round 2

Reviewer 1 Report

The authors have addressed most of the comments that I suggested in the first review. However, I am still concerned about the number of comparisons and how the authors generalize their results while they only find some statistical associations over a great number of possibilities. I would strongly suggest improving the summary of the results (maybe using a figure or a different method than the table). 

Author Response

Point 1: The authors have addressed most of the comments that I suggested in the first review. However, I am still concerned about the number of comparisons and how the authors generalize their results while they only find some statistical associations over a great number of possibilities. I would strongly suggest improving the summary of the results (maybe using a figure or a different method than the table).

Response 1: Thank you for the suggestion. A new figure (attached file here) with description has been added in the Results part to improve the summary of our findings (Page 9, Line 254-257; Page 8, Line 247-252), and the significant associations of built environment attributes with physical activity and physical fitness were listed in the figure. Because Table 4-5 provided more details about the independent association between each built environment attribute and PA/physical fitness, they were still kept in the main text.

This manuscript is a resubmission of an earlier submission. The following is a list of the peer review reports and author responses from that submission.

Round 1

Reviewer 1 Report

Physical activity is fundamental to human health, at the same time a greater focus is being placed on the role of the built environment in promoting physical activity. This is an important topic deserving time and consideration.

Abbreviations used in the text should be explained at first mention (this also applies to the abstract – also BE and PA).

Abstract:

Line 23/24: can we consider a 15-year-old to be an adult?

Introduction:

Line 45: Abbreviations should be explained

Line 53: “To date, the relationship between BE and PA has been confirm by studies in Brazil [15-17], the 53 Czech republic, Poland [18], and some cities in China, such as Shanghai [19], Hangzhou [20]” - Please briefly specify this relationship

Materials

Sample and Study Design

Line 67: “between 15 and 69 years old” It is a large age range - how can authors explain such a choice? Age also has an impact on lifestyle and physical activity.

Results:

Line 118 and table 1: „30.8% aged 16-40 years” - That's not consistent with the description of the study group, which includes 15-69 years old

Table 3, 4, 5 - I have a question about statistical significance, as described in the text? Statistical significance is not marked as described (* indicates significant correlation (P<0.05).

Discussion:

Line 186 – “Results… may be different due to differences in sample characteristics (e.g., children and adolescents, older adults, women)” - The reviewer wonders about the choose a large age range, the authors themselves point this out.

The discussion should refer to the results of other authors. At the moment it is only a description of the results obtained.

Conclusions:

What can the results bring to the main topic?

The comments presented of the manuscript do not diminish its substantive value.

I recommend publishing the manuscript after prior correction of the text or the Authors’ reference to the specific comments.

Reviewer 2 Report

This study aimed to examine associations between the built environment and PA, as well as various health parameters. It is written clearly and concisely with simple and easily understood methodology. However, the issue of multiple comparisons seems quite relevant, especially considering the disparate findings. If, with this many variables, there was some pattern to the findings, the outcomes may have some relevance. However, the findings seem to be random, and the authors’ discuss them as if each is relevant, rather than addressing the fact that they seem unrelated.
Other comments are below.

Lines 60, 61 state that “there was a positive correlation between BE and residents' PA, and a negative correlation between BE and health related fitness in this population with different levels of PA.” But BE isn’t “greater” or “lesser”, it must first be defined. So it isn’t clear what these associations mean.

Line 75: BMI, BFP, and BP were defined in the abstract, but must be defined again the first time they are used within the manuscript.

Line 114: Why did the authors control only for age? Why not include all variables from Table 1?

Table 2: Make the title more descriptive so the table can stand alone. (i.e., explain precisely what each column is showing) And a footnote defining each group would also be helpful.
In the methods PA is grouped into work-related PA, transport PA, domestic and gardening PA, and leisure PA. Table 2 uses different grouping categories.

Lines 124-129: As in the table, explain this in more detail. For example, rather than saying, “In the Moderate-PA groups, the proportion of leisure time PA was the largest, and that of working PA was the smallest.” say something like, “Participants whose overall PA put them in the Moderate-PA group spent the highest proportion of their time in leisure time PA, and the lowest proportion of their time in working PA.”

Line 125: Rather than saying that working PA accounted for the largest proportion of PA in all participants, say “the sample overall.”

Lines 195-198 feel like a non-sequitur.

Reviewer 3 Report

The article entitled “Association of Built Environment with Physical Activity and Health in Chinese Adults Living in Xi'an, China” presents a research study aiming to seek the relationship between different built environment variables, physical activity and health. Despite the interest of this topic, there are major concerns that the authors would need to answer before considering it for further review/publication:

  • There is a potential same source bias when studying the relationship of self-reported built environment and self-reported physical activity. This bias could alter the results of the paper. There are several ways of dealing with this bias ( https://www.sciencedirect.com/science/article/pii/S0213911114000077?via%3Dihub )
  • The paper does not address the novelty of their findings. There are several cross-sectional papers all over the world studying the association between the built environment, physical activity, and health. What’s the novelty of these findings? What can we learn from this study?
  • The authors should consider important confounders. As they set in the statistical analysis section of the paper: “Age was the control variable”. It is worrying that they did not consider any socioeconomic variables, sex, or any other variables as potential confounders of the association
  • The general linear models do not take into account the potential clustering of the data. Mixed models might be a better alternative
  • It is not clear in the methods section how they use physical activity. I understand the classification of high-medium-low is based on METS, but what about purpose-specific physical activity as an outcome?
  • It is not clear why the authors do separate models for different physical activity levels, and if they have done a formal interaction test.
  • It is not clear how and when the questionnaire was designed, what were the other questions, and how the sampling was made.
  • Built environment methods tend to correlate with each other. Where they introduced together in the model? Why there was no test of an overall measure of the built environment?